Outcomes of hospitalized patients with COVID-19 according to level of frailty

Andrés-Esteban Eva María 1 2
http://orcid.org/0000-0003-4852-4148 Quintana-Diaz Manuel 1 3
Ramírez-Cervantes Karen Lizzette 1 4
Benayas-Peña Irene 1
http://orcid.org/0000-0002-6479-4220 Silva-Obregón Alberto 1 5
http://orcid.org/0000-0002-3006-5353 Magallón-Botaya Rosa 6
http://orcid.org/0000-0001-6705-7122 Santolalla-Arnedo Ivan 7
Juárez-Vela Raúl 1 7 raul.juarez@unirioja.es
http://orcid.org/0000-0001-8607-3195 Gea-Caballero Vicente 8 9
1 Grupo PBM, Instituto de Investigación-IdiPaz , Madrid, Madrid , Spain
2 Universidad Rey Juan Carlos , Madrid, Madrid , Spain
3 Servicio de Medicina Intensiva, Hospital Universitario La Paz , Madrid, Madrid , Spain
4 Departamento de Prevención, Asociación Española contra el Cáncer , Madrid, Madrid , Spain
5 Servicio de Medicina Intensiva, Hospital Universitario de Guadalajara , Guadalajara, Guadalajara , Spain
6 Departamento de Medicina, Psiquiatría y Dermatología, Universidad de Zaragoza , Zargoza, Aragón , Spain
7 Universidad de La Rioja, Centro de Investigación Biomédica de La Rioja-CIBIR , Logroño, La Rioja , Spain
8 Nursing School La Fe., Adscript center of Universidad de Valencia. , Valencia, Valencia , Spain
9 Research Group GREIACC, Health Research Institute La Fe. , Valencia, Valencia , Spain
Palazón-Bru Antonio
Electronic publication date: 2021 Apr 13
Publication date: 2021
Volume: 9
Electronic Location ID: e11260
Received 2020 Dec 14; Accepted 2021 Mar 22
Copyright: © 2021 Andrés-Esteban et al.
Copyright year: 2021
Copyright holder: Andrés-Esteban et al.
License: This is an open access article distributed under the terms of the Creative Commons Attribution License, which permits unrestricted use, distribution, reproduction and adaptation in any medium and for any purpose provided that it is properly attributed. For attribution, the original author(s), title, publication source (PeerJ) and either DOI or URL of the article must be cited.
License URL: https://creativecommons.org/licenses/by/4.0/

Keywords: Aged, COVID-19, SARS-CoV-2, Frailty, Mortality, Coronavirus infection, Complications, Prognosis

Funding: ISCIII COV20/00519 and COV20-00634 This research is funded by the ISCIII with code COV20/00519 and COV20-00634. The funders had no role in study design, data collection and analysis, decision to publish, or preparation of the manuscript.

==============================
Background

The complications from coronavirus disease 2019 (COVID-19) have been the subject of study in diverse scientific reports. However, many aspects that influence the prognosis of the disease are still unknown, such as frailty, which inherently reduces resistance to disease and makes people more vulnerable. This study aimed to explore the complications of COVID-19 in patients admitted to a third-level hospital and to evaluate the relationship between these complications and frailty.

Methods

An observational, descriptive, prospective study was performed in 2020. A sample of 254 patients from a database of 3,112 patients admitted to a high-level hospital in Madrid, Spain was analyzed. To assess frailty (independent variable) the Clinical Frailty Scale (CFS) was used. The outcome variables were sociodemographic and clinical, which included complications, length of stay, intensive care unit (ICU) admission and prognosis.

Results

A total of 13.39% of the patients were pre-frail and 17.32% were frail. Frail individuals had a shorter hospital stay, less ICU admission, higher mortality and delirium, with statistical significance.

Conclusion

Frailty assessment is a crucial approach in patients with COVID-19, given a higher mortality rate has been demonstrated amongst frail patients. The CFS could be a predictor of mortality in COVID-19.

Introduction

The novel severe acute respiratory syndrome coronavirus 2 (SARS-CoV-2) and the consequential coronavirus disease 2019 (COVID-19) have been detected in more than 99.6 million people worldwide and have caused more than 2.1 million deaths. As of 24 January 2020, 2.6 million cases had been diagnosed in Spain and 55,441 people had died from the disease (Worldmeters, 2020).

In Spain most cases of COVID-19 have occurred in people aged 60 or above and fatality rates have showed to rise exponentially with age. In addition, the hospitalized people aged >80 account for 28.7% of admissions, reaching the highest mortality; in Italy (one of the countries with the highest infection and mortality rates in the world), until July, mortality in the population group >80 years was as high as almost 60% (Sornette et al., 2020). In the elderly, the presence of comorbid conditions has shown significant negative effects on prognosis (Maltese et al., 2020). In Europe, the first large cohort of hospitalized patients showed that older age and the presence of comorbidities were more common among subjects with fatal outcomes, both for the entire cohort and for those admitted to the Intensive Care Units (ICUs). However, the patients admitted to the ICU were older, had a higher male/female ratio and a higher prevalence of hypertension, obesity, diabetes mellitus and chronic obstructive pulmonary disease (Borobia et al., 2020).

Despite the presence of comorbidities increases the likelihood of death among patients with COVID-19 (RR (95% confidence interval (CI) = 1.69)) (Abate et al., 2020); other characteristics such as frailty might predict poor prognosis (Hewitt et al., 2020). Frailty has been defined as “a medical syndrome with multiple causes and contributors that is characterized by diminished strength, endurance, and reduced physiologic function that increases an individual’s vulnerability for developing increased dependency and/or death” (Morley et al., 2013). Despite it has been related to disability, agedness and multimorbidity, they are not synonymous. For instance, frailty is more prevalent in older people, however, it can also affect the middle-aged (Xue, 2011; Chong et al., 2020).

Several studies have described frailty as a poor outcome predictor in illness (Morley et al., 2013), particularly in ICU patients (So et al., 2018; Muscedere et al., 2017; Bellelli et al., 2020). It has also been associated with more ICU admissions, higher in-hospital and ICU mortality rates, greater disability, and increased use of all forms of organ support (e.g., mechanical ventilation, noninvasive ventilation, vasopressors) (Zampieri et al., 2018). In the context of the current pandemic, a recent study has shown that frailty may account for as much as 49.4% of hospital COVID-19 admissions (Hewitt et al., 2020; Zampieri et al., 2018) and higher mortality rates and hospital stays have been described (Maltese et al., 2020). We have documented mortality that is clearly associated with age, rising drastically after 65 years of age, with an average of between 76 and 80 years of age (approximately) (Sornette et al., 2020).

Stratifying patients according to their level of frailty is key to determine the risks to which they are exposed (Changfeng, Shouyan & Yu, 2019). Currently, a gold standard instrument for assessing frailty is lacking; however, the International Conference of Frailty and Sarcopenia Research Task Force recommends using a validated setting-specific instrument to screen frail elderly adults (Dent et al., 2019; Hoogendijk et al., 2019), such as the Clinical Frailty Scale (CFS).

In the United Kingdom (UK), the COVID-19 rapid action guide from the National Institute for Clinical Excellence (NICE) recommends the use of the CFS to assess frailty according to patient groups (NICE Guideline, 2020). This has generated controversy among some specialists, who argue that decisions should be made on an individual basis (Hewitt et al., 2020). However, frailty assessments employing the CFS has resulted to be an ai to decision-making regarding resources and patient services (Cesari, 2019). For instance, in primary care setting, frailty screening, together with the right assessment of illness severity could be beneficial for the right allocation of patients avoiding excessive hospitalizations (Maltese et al., 2020). In addition, identifying frail and pre-frail individuals could improve the clinical management of this condition by focusing on the exhaustive monitoring and long recovery and rehabilitation that frail individuals may require (Hewitt et al., 2020). On the other hand, in the context of COVID-19 infection, the assessment of frailty could be a better determining factor for decision-making than age (Hewitt et al., 2020). Therefore, the rationale for assessing frailty in COVID-19 individuals is justified (Maltese et al., 2020).

Given that frailty could associate a high risk of complications, severe manifestations and death in COVID-19 patients, our study aimed at exploring the outcomes of patients admitted to a third-level hospital with COVID-19 according to the degree of frailty and to study the association between frailty and death. We hypothesize that, as previously reported, frail individuals will experience poorer health outcomes.

Materials and Methods

Design

We performed an observational, descriptive and retrospective

Population and sample

A total of 3,581 clinical episodes were extracted from a database of 3,112 patients from La Paz University Hospital (LPUH) during the period between 15 July 2020 and 31 July 2020, meaning a patient could have been admitted to the hospital several times. The database included sociodemographic data, clinical status, laboratory findings, and clinical management of the patients admitted with a respiratory infection by SARS-CoV-2 (determined by polymerase chain reaction) since the beginning of the current pandemic. A sample of 254 patients was randomly chosen from the total population, and with this sample size, in terms of the most frequent complications, a test power of at least 0.8744 was obtained.

Frailty assessment

Frailty was assessed by the CFS, which is a tool initially developed by the Canadian Health Study of Aging based on the Frailty Index (FI) to be used in patients older than 65 years. It was created with the objective of developing a simple instrument to stratify patients according to their degree of vulnerability and to relate it to patients’ needs and prognoses. However, contrary to the FI, that includes comorbidity within its score, which has been independently associated with worse clinical outcomes in COVID-19 patients, the CFS does not take into account comorbidities for its assessment, allowing to focus on the investigation of the accurate relationship between frailty and clinical outcomes (Kow & Hasan, 2020).

The CFS is a practical, efficient tool that assigns a score between 1 and 7 based on activity, functionality and disability (Rockwood et al., 2005). It classifies patients as robust (score 1–3), pre-frail (score 4), and frail (score >5) (Dent et al., 2019). This stratification allowed to identify not only frail individuals, who usually have higher risk of serious health complications and worse prognosis; but pre-frail subjects as well, who have reported an increased risk of negative events, such as an elevated mortality from all causes (Changfeng, Shouyan & Yu, 2019).

The CFS scale was not taken directly from the medical record but involved analyzing family medical history, nursing history, and in some cases interviewing the patient directly. Six doctors spent 15 days collecting this information and classifying patients according to the CFS scale. Parallelly, a group of experts trained in the use of frailty scales classified the same patients according to their level of frailty obtaining a Kappa 0,89 (95% CI [0.82–0.94])

Variables

Exposure variables

Frailty as measured by CFS. According to this scale, the levels of frailty range from robust (score 1), to those who are completely dependent on others (score 7), to terminal patients (score 9). A score of 5 or more is indicative of frailty, and a score of 4 is considered a pre-frailty state. Patients scoring 1–3 are considered healthy or robust (Rockwood et al., 2005).

Covariates (Secondary variables)

The data regarding secondary variables included demographic information (age, gender); previous medical history and comorbidities; clinical variables (from arrival at the emergency room to hospital admission); laboratory test findings; and diagnostic imaging performance and results.

Outcome variables

The main variables of the study were based on patients’ outcomes (e.g., the presence of complications, need for ICU admission, days of stay, mortality) and on a frailty assessment using the CFS.

Statistical analysis

Quantitative variables were described using robust statistics, such as median and interquartile interval, whereas for qualitative variables a frequency distribution was used. For the comparison of quantitative not normally distributed variables between frailty groups, the Kruskal–Wallis non-parametric H test was used, based on the Shapiro–Wilk test. Finally, for the comparison of qualitative variables, the chi-squared test was used.

The survival estimate was made using the Kaplan–Meier method comparing the survival curve between groups with the log-rank test. The multivariate analysis was carried out by means of Cox regression, with the forward conditional method, introducing as independent variables the CFS scale and the variables that obtained statistical significance in the bivariate analysis or that could have a clinically plausible implication. The results of the multivariate model were presented as a Hazard ratio (95% CI).

Statistical analysis was performed with STATA v16.0 and a pvalue of 5% was considered statistically significant.

Ethical considerations

The study was performed in accordance with the principles of the Declaration of Helsinki (2008 update; available on the World Medical Association website-http://www.wma.net/e/policy/b3.htm) and in accordance with the standards of good clinical practice as described in the ICH Harmonized Tripartite Guidelines for Good Clinical Practice (2001), and the guidelines for Good Epidemiological Practice (http://www.ieaweb.org/GEP07.htm). The study was approved by the Clinical Research Ethics Committee of LPUH, Madrid, with LPUH code: PI-4155. It was not necessary a formulary of informed consent due to the anonymized database was extracted.

This study was conducted in accordance with European and Spanish regulations for the protection of personal data (Organic Law 3/2008).

Results

Patients characteristics

Complete information for all variables was collected from 254 patients. According to the CFS instrument 69.29% were classified as robust (CFS = 1–3), 13.39% as pre-frail (CFS = 4) and 17.32% as frail (CFS > 5). Table 1 shows demographic and housing characteristics, COVID-19 contagion risks, level of dependency and previous comorbidities of the patients studied according to their level of frailty.

Table 1 Demographic and housing characteristics, COVID-19 contagion risks, level of dependency and previous comorbidities according to level of frailty.

Variable	CFS	p-Value	
Robust (CFS ≤ 3)	Pre-frail (CFS = 4)	Frail (CFS ≥ 5)	
	176 (69.29%)	34 (13.39%)	44 (17.32%)		
Sex				0.018	
Men	113 (69.94%)	23 (69.70%)	19 (43.18%)		
Women	61 (35.06%)	10 (30.30%)	25 (56.82%)		
Age (median, IQI)	69.50 (55.00–79.00)	76.00 (66.50–82.50)	81.50 (72.00–87.75)	<0.001	
Smoker	13 (7.39%)	3 (8.82%)	3 (6.82%)	0.942	
Health care provider	9 (5.77%)	1 (3.23%)	0 (0.00%)	0.260	
Housing conditions					
				<0.001	
Not overcrowding	170 (97.70%)	32 (96.97%)	32 (72.73%)		
Nursing home	3 (1.72%)	1 (3.03%)	12 (27.27%)		
Shelter	1 (0.57%)	0 (0.00%)	0 (0.00%)		
COVID-19 contagion risks					
Direct/close contact with an individual with COVID-19	21 (13.91%)	3 (10.34%)	7 (17.95%)	0.665	
Nosocomial infection	19 (11.11%)	7 (21.88%)	14 (31.82%)	0.003	
Level of dependency					
				<0.001	
Dependent	3 (1.89%)	1 (3.23%)	13 (30.95%)		
Semi-dependent	1 (0.63%)	3 (9.68%)	5 (11.90%)		
Independent	155 (97.48%)	27 (87.10%)	24 (57.14%)		
Comorbidities					
Heart disease	33 (18.75 %)	14 (41.18%)	18 (40.91%)	0.001	
High blood pressure	85 (48.30%)	19 (55.88%)	30 (68.18%)	0.057	
Chronic obstructive pulmonary disease (COPD)	14 (7.95%)	6 (17.65%)	5 (11.36%)	0.206	
-Chronic bronchitis	12 (6.82%)	4 (11.76%)	10 (22.73%)	0.007	
Asthma	9 (5.11%)	4 (11.76%)	2 (4.55%)	0.294	
Chronic kidney disease	14 (7.95%)	7 (20.95%)	9 (20.45%)	0.017	
Moderate kidney damage	1 (0.57%)	2 (5.88%)	6 (13.64%)	<0.001	
Mild liver disease	7 (3.98%)	0 (0.00%)	0 (0.00%)	0.203	
Chronic neurological disorder	9 (5.11%)	7 (20.95%)	9 (20.45%)	0.001	
Malignant neoplasm	21 (11.93%)	6 (17.65%)	9 (20.45%)	0.288	
Chronic hematological disease	11 (6.25%)	6(17.65%)	4 (9.09%)	0.085	
Obesity	20 (11.36%)	10 (29.41%)	6 (13.64%)	0.022	
Diabetes	40 (22.73%)	10 (29.41%)	16 (36.36%)	0.162	
Rheumatological disorder	21 (11.93%)	4 (11.76%)	8 (18.18%)	0.530	
Dementia	1 (0.57%)	1 (2.94%)	15 (34.09%)	<0.001	
Dyslipidemia	73 (41.48%)	18 (52.94%)	25 (56.82%)	0.124	
Common mental disorder (anxiety, depression)	17 (9.66%)	4 (11.76%)	10 (22.73%)	0.060	
Severe mental disorder (schizophrenia, bipolar disorder, other)	1 (0.57%)	0 (0.00%)	1 (2.27%)	0.445	

The individuals classified as frail were mostly women (56.82%), with an older age (median (IQR): 81.5 (72–87.75)) compared with the robust (median (IQR): 69.50 (55.00–79.00)) and pre-fail (median (IQR): 76.00 (66.50–82.50)) subjects. Frail patients had a higher prevalence of suspected nosocomial transmission (27.27%), compared with robust (1.72%) or prefrail (3.03%) individuals.

In Table 1 it can also be observed that most comorbidities had a similar distribution among the three groups defined above, except for heart disease—for which the robust group had a very low prevalence compared with the pre-frail and frail individuals (18.75% vs. 41.18% and 40.91%, respectively; p = 0.001)—and chronic bronchitis, for which the prevalence grew linearly among the three groups (p = 0.007). With the same pattern as heart disease, chronic renal disease was less prevalent among robust patients (7.95%) compared to pre-frail and frail groups (20.95% and 20.45%, respectively; p = 0.017). Differences were also observed regarding the prevalence of previous neurological disorders, obesity and dementia.

At arrival to the emergency room the clinical presentation of COVID-19 was similar among groups; however, as shown in Table 2, the highest incidence of altered behavior and level of consciousness occurred in frail patients (11.93%, p =< 0.001)

Table 2 Clinical symtomps at arrival to the emergency department according to level of frailty.

Symptoms and evolution	Clinical frailty scale (CFS)	p-Value	
Robust (CFS ≤ 3)	Pre-frail (CFS = 4)	Frail (CFS ≥ 5)		
Fever	140 (79.55%)	31 (91.18%)	31 (72.73%)	0.128	
Headache	21 (11.93%)	5 (14.71%)	4 (9.09%)	0.745	
General discomfort	76 (43.18%)	8 (23.53%)	21 (47.73%)	0.066	
Myalgia	38 (21.59%)	7 (20.59%)	3 (6.82%)	0.079	
Rhinorrhea	9 (5.11%)	3 (8.82%)	0 (0.00%)	0.173	
Dysgeusia	2 (1.14%)	0 (0.00%)	1 (2.27%)	0.651	
Anosmia	1 (0.57%)	0 (0.00%)	0 (0.00%)	0.801	
Cough	110 (62.50%)	20 (58.82%)	20 (45.45%)	0.121	
Productive cough	40 (22.73%)	12 (35.29%)	10 (22.73%)	0.284	
Odynophagia	12 (6.82%)	1 (2.94%)	0 (0.00%)	0.153	
Chest pain	15 (8.52%)	4 (11.76%)	3 (6.82%)	0.738	
Costal pain	6 (3.41%)	1 (2.94%)	0 (0.00%)	0.465	
Hemoptoic expectoration	1 (0.57%)	0 (0.00%)	2 (4.55%)	0.073	
Dyspnea	89 (50.57%)	16 (47.06%)	25 (56.82%)	0.665	
Abdominal pain	16 (9.09%)	2 (5.88%)	5 (11.36%)	0.704	
Diarrhea	36 (20.45%)	5 (14.71%)	6 (13.64%)	0.482	
Nausea	22 (12.50%)	2 (5.88%)	4 (9.09%)	0.478	
Vomiting	7 (3.98%)	2 (5.88%)	2 (4.55%)	0.880	
Altered level of consciousness*	21 (11.93%)	2 (5.88%)	14 (31.82%)	<0.001	
Altered behavior	6 (3.41%)	0 (0.00%)	7 (15.91%)	<0.001	
Notes:

* The assessment of level of consciousness and behavior was performed by a medical team at the emergency room.

If necessary, further information was obtained from the person accompanying the patient at arrival.

These data were gathered from medical records.

COVID-19 outcomes

Table 3 shows all the in-hospital complications related to COVID-19 according to level of frailty. The most common complications among patients with COVID-19 were acute renal failure (46 patients), delirium (38 patients) and acute respiratory distress syndrome (ARDS) (51 patients).

Table 3 Complications observed during hospital admission according to level of frailty.

Complications	Clinical frailty scale (CFS)	p-Value	
Robust (CFS ≤ 3)	Pre-frail CFS = 4)	Frail (CFS ≥ 5)	
Noncomplicated disease	32 (18.18%)	4 (11.76%)	8 (18.18%)	0.655	
Mild pneumonia	65 (39.93%)	8 (23.54%)	8 (18.18%)	0.031	
Severe pneumonia	84 (47.73%)	21 (61.76%)	29 (65.91%)	0.051	
Acute Respiratory Distress Syndrome (ARDS)	35 (19.89%)	8 (23.53%)	8 (18.18%)	0.837	
Myocarditis	1 (0.57%)	0 (0.00%)	0 (0.00%)	0.801	
Pericarditis	1 (0.57%)	0 (0.00%)	0 (0.00%)	0.801	
Arrhythmia	6 (3.41%)	2 (5.88%)	2 (4.55%)	0.774	
Cardiac ischemia	2 (1.14%)	0 (0.00%)	0 (0.00%)	0.640	
Cardiac arrest	12 (6.82%)	0 (0.00%)	4 (9.09%)	0.229	
Bacteremia	14 (7.95%)	2 (5.88%)	4 (9.09%)	0.871	
Coagulation disorder	17 (9.66%)	5 (14.71%)	3 (6.82%)	0.505	
Subsidiary anemia	7 (3.98%)	1 (2.94%)	2 (4.55%)	0.936	
Rhabdomyolysis	4 (2.27%)	0 (0.00%)	1 (2.27%)	0.674	
Acute kidney failure	28 (15.91%)	6 (17.65%)	12 (27.27%)	0.215	
Pancreatitis	0 (0.00%)	0 (0.00%)	1 (2.27%)	0.091	
Liver failure	6 (3.41%)	0 (0.00%)	2 (4.55%)	0.490	
Delirium	15 (8.52%)	4 (11.76%)	19 (43.18%)	<0.001	
Psychiatric complications	6 (3.41%)	1 (2.94%)	1 (2.27%)	0.926	
Adverse drug reaction	15 (8.52%)	5 (14.71%)	1 (2.27%)	0.138	
Serious adverse drug reaction	2 (1.14%)	1 (2.94%)	1 (2.27%)	0.682	
Sepsis	7 (3.98%)	3 (8.82%)	3 (6.82%)	0.429	
Septic shock	12 (6.82%)	2 (5.88%)	3 (6.82%)	0.980	
Intensive care unit admission	55 (31.25%)	13 (38.24%)	5 (11.36%)	0.014	
Death	62 (35.23%)	16 (47.06%)	26 (59.09%)	0.012	
Length of hospital stay (LOS) (median; IQR)	11 (9–19)	10 (7–14)	5 (1–11)	0.296	
LOS of survivors (median; IQR)	12 (17–23)	14 (18–27)	10 (13–28)	0.006	

Despite the length of hospital stay was similar between groups (p = 0.296), ICU admissions (p = 0.014) were less prevalent among frail individuals. Moreover, mortality rates increased linearly among the three groups (p = 0.012).

A statistically significant difference was found regarding delirium, whose incidence was 8.52% in robust patients, 11.76% in pre-frail subjects and 43.18% in frail individuals. Delirium was also associated with a higher mortality rate, being presented in 27.30% of frail patients versus in 4.55% without it; p < 0.001.

No differences were found between groups regarding the rest of the COVID-19 outcomes shown in Table 3. However, it should be noticed that heart failure appeared as a complication of the disease in 5.88% and 6.82% of pre-frail and frail individuals, respectively, compared with an incidence of 1.14% in the robust group; nonetheless, this difference did not reach statistical significance (p = 0.059).

Figure 1 shows the Kaplan–Meier survival estimation curves for each group of patients, according to the CFS frailty scale. In this graph a huge difference during the first fortnight can be detected, especially for patients identified as frail. The differences observed were statistically significant (p value < 0.001).

Figure 1 Overall survival.

Finally, a multivariate model was performed using a Cox regression model from the variables shown in the Table 4. The results demonstrated that frailty was associated with mortality (HR: 1.39, 95% CI [1.07–1.81]). However, other factors such as chronic heart disease, chronic kidney disease, and high blood pressure were also associated with death.

Table 4 Cox regression according to CFS*.

			CI (95%)	
HR	p Value	Lower	Upper	
Frail vs. pre-frail/robust	1.39	0.014	1.07	1.81	
Chronic heart disease	1.68	0.035	1.04	2.72	
Chronic kidney disease	2.08	0.018	1.13	3.81	
Hypertension	1.51	0.049	1.00	2.38	
C—Harrell = 0.681					
Note:

* Sex, age, overcrowding, nosocomial infection, dependency, heart disease, COPD, asthma, kidney disease, neurological disorder, obesity, dementia, ICU admission, hypertension, mortality and frailty were the variables included in the multivariable model.

Discussion

Our study introduces important findings. In this retrospective cohort study, we found that frailty was a major predictor of mortality among hospitalized elderly patients with COVID-19; cardiovascular disease, chronic kidney disease, and hypertension were also shown to predict decreased survival in the fully-adjusted model of frailty.

By reporting the prevalence of the complications in a sample of 254 patients admitted with a diagnosis of COVID-19, we have contributed to increase the current knowledge regarding this new disease. The main findings of our study focused, on one hand, on the sociodemographic characteristics of individuals with COVID-19; and on the other hand, on the clinical symptoms at admission and in-hospital complications of these patients stratified by frailty profiles. Among the highlights of our clinical findings are the presentation of behavioral alterations and delirium, the higher mortality rate and the less ICU admission among frail individuals.

Sociodemographic characteristics of interest

In our study, we found that 17.32% of the patients were classified as frail, however, if the subjects with pre-frailty were considered, the total prevalence would increase to 30.71%, which is still a much lower result compared with another study that have reported proportions of 39.6–42.0% (Hewitt et al., 2020). Choi, Ahn & Kim (2015) demonstrated that the most affected frail patients were women and residents of nursing homes, which could be related to the advanced age and the dependency situation that feature these patients. In our study, old age, female sex, dependency, living in a nursing home, and having comorbidities were the high-risk profiles of frail individuals. These data are relevant because frailty is a condition that increases the risk of hospitalization, unwanted incidents and mortality (Martínez-Reig et al., 2016).

The devastating effects of the pandemic on nursing homes are well known causing a disproportionate number of infections and deaths worldwide (Rutten et al., 2020). The typical response to prevent the transmission of the virus in nursing homes has been the aggressive isolation of patients, sometimes ignoring its possible collateral effects in patients with dementia, delirium, or other mental health problems, for whom human relationships are essential (Tan & Seetharaman, 2020). It has been reported that in similar previous epidemics (SARS-CoV in 2003 and MERS-CoV in 2012) approximately 35% of the survivors showed psychiatric disorders during their early recovery, including depression, stress and anxiety, in addition to other minor symptoms (Mak et al., 2009).

The preexisting medical conditions that showed significant associations with frailty in our study were kidney disease, dementia, cardiovascular and respiratory diseases; therefore, the patients with COVID-19 could fit the typical profile of frail individuals: women, elderly, many of them institutionalized in nursing homes, and with comorbidities (Romero Cabrera, 2011). Comorbid conditions are also a notable element as a prognostic factor for the disease; some studies have especially highlighted advanced age, frailty, and the prior presence of chronic diseases (comorbidity) as important risk factors in COVID-19 (Landi et al., 2020). Hence, the control of comorbidities could be a key measure to improve COVID-19 outcomes, especially considering that a large proportion of deaths occurred during influenza pandemics were due to a decompensation of previous medical conditions and their subsequent complications, which is actually being reported during the current pandemic (Garnier-Crussard et al., 2020).

Clinical symptoms and complications

Although CFS does not consider multimorbidity within its classification, it is known that they are related to worse outcomes. Our results showed a similar distribution of multimorbidity (without prioritizing any disease over another) among the three groups with some exceptions: neurological disease (20.45%), dementia (34.09%), heart disease (40.91%), chronic bronchitis (22.73%) and obesity (13.64%) that were more prevalent among frail patients.

Some of the chronic pathologies that were associated with poor prognosis were chronic heart disease, chronic kidney disease and hypertension, which are frequent conditions among elderly and frail people that increase their vulnerability to suffering worse outcomes.

The relationship between frailty/multimorbidity, in view of the results, since multimorbidity and frailty may be bidirectionally related, should lead us to a deep reflection about discuss the role of a multidimensional approach integrating multimorbidity and frailty in the management of hospitalized patients (Boeckxstaens et al., 2015). As Price et al. (2020) argue in their research protocol, it is important to know the association between multimorbidity and frailty with health outcomes, as this could enable better allocation of specialized resources, and especially holistic and shared decision making with patients.

It is well known that respiratory viruses, such as coronaviruses, can produce neuroinvasion and neurological symptoms, in addition to the severe hypoxia that respiratory symptoms can generate, especially in the elderly (Carod Artal, 2020). According to our results, neurological complications from COVID-19 were uncertain, with nonspecific and systemic symptoms (headaches, dizziness, etc.), highlighting the most common and more specific disorders of smell and taste. However, the most remarkable results observed in frail people were alterations of behavior and delirium (43.18%). Mao et al. established that alterations in the level of consciousness can be severe in almost 15% of cases, with only 2.4% considered mild (Mao et al., 2020). However, our results were identical to Mcloughlin et al. (2020) (42%) and higher than that reported by Zazzara et al. (2020) (38%). Hence, an episode of delirium in elderly and frail patients could be the first symptom of COVID-19, as suggested by another author (O’Hanlon & Inouye, 2020). In addition, since frailty implies physical and mental deterioration, these data highlight the importance of including sociopsychological support in our clinical management in order to reduce fatal outcomes among frail patients.

Frailty has been considered a predictor of mortality (Changfeng, Shouyan & Yu, 2019; Dent et al., 2019). In our study, in addition to confirming the relationship between frailty and mortality in patients with COVID-19 (HR: 1.39, 95% CI [1.07–1.81]), as previously hypothesized, delirium behaved as a predictor of death in frail people. Despite that another study (Mao et al., 2020) did not find an association between delirium and mortality, this complication is being studied and some studies have already offered clinical guidance for treating delirium in patients with COVID-19 in ICUs (Kotfis et al., 2020).

We also found, with statistical significance, a lower tendency for COVID-19 pneumonia to be mild in frail people. However, without statistical significance, a greater tendency for pneumonia to be severe in the groups of frail and pre-frail individuals was observed (in more than 60% of the patients of both groups pneumonia was severe, while in the robust group it occurred in 47.73% of the cases). These findings are concerning, given SARS-CoV-2 appears to be a potentially similar virus to SARS-CoV and MERS-CoV-6; in which, elderly patients had a greater tendency to develop serious respiratory diseases, greater complications, more disability and higher mortality (Garnier-Crussard et al., 2020). However, these clinical overviews are similar to the findings of our study, which demonstrated an increased risk of developing severe pneumonia and a high risk of mortality for the elderly, frail, and those with multimorbidity (Weiss & Murdoch, 2020).

A previous study (Miles et al., 2020) has suggested that frailty might not be a good prognostic factor for death; however, our estimations showed a remarkable reduction in survival rates in frail patients which was not observed in prefrail and robust individuals. Other remarkable findings were the significant difference observed regarding the need for ICU admission among frail subjects, and its association with mortality. It was concerning to find that the subgroup of frail people had a mortality rate close to 60%, but only 11.36% were admitted to the ICU. Lower rates were obtained by Hewitt et al. (2020) who reported a mortality rate close to 50% in frail people from the UK, with 7% of terminal patients. However, it must be taken into consideration that during the initial period of the pandemic the burden of hospital strains triggered protocols of limited ICU admissions that discouraged patients who were not considered for resuscitation, i.e., patients with a poor prognosis; therefore, frail individuals could have also been discouraged from ICUs. In addition, due to frail subjects were older than pre-frail and robust individuals, it suggests that age could have been a determining factor in the decision-making of COVID-19 treatment and health resource allocation. Nevertheless, a recent study has concluded that even severely frail elderly patients with a poor prognosis can be benefited from hospitalization for COVID-19, and their survival could increase if enough resources were available for them (De Smet et al., 2020); therefore, the risk of worse outcomes and less ICU admission among the elderly could be better explored to avoid ageism in a setting of limited health resources.

This serious situation is not exclusive to Spain. When COVID-19 hospital admissions exceeded the availability of beds, hospitals in China and Italy (among other countries in Europe) also had to face the difficult task of classifying patients in order to allocate resources (Boreskie, Boreskie & Melady, 2020). Age was an important criterion in the triage of limited resources, as we observed (Boreskie, Boreskie & Melady, 2020); for example, in Italy, generating some rejection by equating old age with frailty (Rosenbaum, 2020). It is remarkable that this situation has affected countries with a very advanced life expectancy, such as Italy or Spain. In the UK, the criterion recommended by the NICE COVID-19 Quick Guide was the assessment of income frailty as a criterion to assist in decision-making in the allocation of resources, with less attention to biological age (NICE Guideline, 2020).

Assessing the level of frailty as a regulatory condition for the allocation of health resources is not new. Some economic studies have previously addressed this problem in response to the high demand for care of frail, elderly, and multimorbidity patients (Cesari et al., 2016). Therefore, it should not be surprising that when health services are strained - such as during the peaks in the COVID-19 pandemic- some patients are ruled out for ICU admission, as seen in our study and in other countries.

Expanding our knowledge about frailty in COVID-19 is essential. The relationship between frailty and mortality, ICU admission, and hospital stay appears clear. However, as a recently published in a systematic review, research should now be directed toward interventions and treatments, given the guidelines for care of frail patients with COVID-19 are not based on strong evidence. As previously addressed, the implications of COVID-19 for the physical and mental health of patients and health professionals should lead us to reflect on the need for developing specific care protocols, based on solid evidence, to facilitate optimal decision-making in vulnerable and frail individuals, especially in terms of the detection of early symptoms for the activation of healthcare processes (Ramírez-Ortiz et al., 2020)

Limitations

Given that our work was a descriptive observational study carried out in a random sample selected from a larger cohort of COVID-19 indivduals, we cannot demonstrate causality in the relationship between the variables studied. In addition, due to the CFS was determined by the revision of medical records there could be slight variability in the assessment when it is performed in a medical setting.

Conclusions

Frailty has shown to worsen the prognosis of COVID-19 patients. Higher frailty scores were associated with an increased mortality rate in hospitalized patients with COVID-19. The presence of delirium in elderly and/or frail patients could be a frequent atypical manifestation of COVID-19 that could also predict a higher mortality rate. The assessment of frailty in hospital settings could be an efficient tool to avoid ageism, improve clinical decision-making and health resource allocation, especially due to frail individuals might require closer monitoring and/or more intensive treatment for their preexisting conditions. Due to frailty may affect both physical, physiological and cognitive functions, frailty itself demands implementation of a holistic approach to patient care.

Supplemental Information

Supplemental Information 1 Code Book.

This codebook converts numbers to their respective factors.

Click here for additional data file.

Supplemental Information 2 Raw Data.

Click here for additional data file.

We thank the COVID@HULP Working Group of Hospital La Paz, Silvia Erce, Raquel Carmona and Aitana López, for their help and support in this investigation.

Additional Information and Declarations

Competing Interests

Author Contributions

Human Ethics

Data Availability

The authors declare that they have no competing interests.

Eva María Andrés-Esteban conceived and designed the experiments, analyzed the data, prepared figures and/or tables, and approved the final draft.

Manuel Quintana-Diaz performed the experiments, prepared figures and/or tables, and approved the final draft.

Karen Lizzette Ramírez-Cervantes conceived and designed the experiments, analyzed the data, authored or reviewed drafts of the paper, and approved the final draft.

Irene Benayas-Peña performed the experiments, analyzed the data, authored or reviewed drafts of the paper, and approved the final draft.

Alberto Silva-Obregón conceived and designed the experiments, authored or reviewed drafts of the paper, and approved the final draft.

Rosa Magallón-Botaya performed the experiments, authored or reviewed drafts of the paper, and approved the final draft.

Ivan Santolalla-Arnedo analyzed the data, prepared figures and/or tables, and approved the final draft.

Raúl Juárez-Vela conceived and designed the experiments, performed the experiments, prepared figures and/or tables, and approved the final draft.

Vicente Gea-Caballero performed the experiments, analyzed the data, prepared figures and/or tables, and approved the final draft.

The following information was supplied relating to ethical approvals (i.e., approving body and any reference numbers):

The study was approved by the Clinical Research Ethics Committee of La Paz University Hospital, Madrid (LPUH code: PI-4155).

The following information was supplied regarding data availability:

The raw data and codebook are available in the Supplemental Files.

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
