# Peer review of "Outcomes of hospitalized patients with COVID-19 according to level of frailty"

_PeerJ, doi:10.7717/peerj.11260_

## Round 0.1 · original submission · Major Revisions

The reviewers have indicated scientific merit in your work. However, there are some issues which you must address in a revised version of the text. Please, see their comments below so as to have more information.

Reviewer 1 ·

Basic reporting

The authors carry out an adequate review of the bibliography for the introduction and part of the results, but they need to explore and discuss their information further. You need to improve figure 1 and review the final analysis of the study.

Experimental design

Method
The authors need to review the type of study, as it is not prospective.
The scale used in the study could be used as the authors put it in the study that is to say without evaluating the elderly and only by the medical records? Does the scale validation for Spanish from Spain indicate this? In the creation of the scale, the authors who created it refer that the scale can be used in this way?
Has the scale been validated for Spanish in Spain? especially the cutoff point? this information does not appear.
The authors need to better explain the instrument used, the extraction of information from medical records.

Validity of the findings

the results need to answer the objective of the study, the authors want to present important data but that in the course of these and the analyzes are lost, leaving aside its main outcome, which is fragility and not the survival of the elderly.
Results
Table 1 not only demographic data appear, but also clinical data (diseases), there is some information with a data that can direct the statistical significance of the category. Review the title of the table 1, it is not adequate according to the data it presents.
In Table 2, presentation of signs and symptoms it is not clear how the change in consciousness and behavior was assessed, making this information more clear.
In table 3, what does identified microorganism mean? Is related to what?
The information about heart disease that the authors refer does not appear in table 3, making this information clearer.
To improve the information in figure 1, it is not recommended to make a figure in 3D. Now, why do figure and not table? What do the authors want to present? The way the data is distributed the information is not adequately evidenced
This information is about what? “Lung disease: bacterial pneumonia, ARDS, pneumothorax, pleural effusion. Neurological disease: meningitis, seizures, stroke. Heart disease: congestive heart failure, myocarditis, pericarditis, endocarditis, arrhythmia, cardiac ischemia, cardiac arrest. Hematological disease: coagulation disorder, subsidiary anemia. Muscle disease: rhabdomyolysis. Digestive disease: digestive bleeding, pancreatitis, liver failure. Psychiatric disease: acute confusion syndrome, psychiatric complications. Infectious complications: infectious upon admission, bacteremia. Inflammatory disease: endocarditis, myocarditis, pericarditis, pancreatitis, meningitis”
Table 4 shows linear regression related to mortality, interesting information but which does not answer the objective of the study. In this sense, the dependent variable, frailty, was considered as independent, which leads to another study objective, to review this information.

Additional comments

Title
The title is in agreement with the objective of the study but not with the results. it is suggested to modify it or review the study results

Abstract
The authors need to improve the method, results and conclusion. It does not adequately answer the study objective. The descriptors used are adequate

Introduction
The researchers describe the situation that Spain experienced with the beginning of the pandemic of COVID-19; bring important data that must be rigorously evaluated, especially with the large number of elderly people who died from the disease or complications from the disease.
At present there is no standard definition of frailty syndrome, the authors bring a definition of it but is it the theoretical framework used in the research? I think they could bring the consensus of Morley's frailty that is widely used and the concept of the instrument used to identify frailty in the elderly in this study.
The authors describe the frailty scale in the introduction but that this should be in the method.
The next paragraph: “In addition, our research group has also considered the CFS as the best tool to use in this pandemic context because it is less time consuming and because ICU patients are often unable to meaningfully participate in frailty assessment” is a conclusion. The authors need to review its location within the text in the corresponding section.
The study needs a plausible justification for why it is important to do it.
The objective of the study is “to explore the clinical complications and prognosis of individuals with COVID-19 admitted to a third-level hospital and to assess the relationship between these complications and prognosis with frailty” but in the introduction, the authors do not write nothing about the complications and prognosis.

Discussion
The authors discuss data on elderly people who live in nursing homes but there is no way to raise this hypothesis because it is data that has not been presented, which can lead the discussion to error. The authors must also inform data on the number of elderly people who are dying in the world, there is already literature on the subject, review the discussion of sociodemographic data.
Contamination by COVID-19 leads to changes in the organism. The authors need to be careful when stating that "an episode of delirium in elderly and fragile patients should lead us to suspect COVID-19," since a deeper evaluation is important for this. It is recommended to revise the indicated paragraph.
The authors refer "in addition to confirming the relationship between frailty and mortality in patients with COVID-19, delirium behaved as a predictor of death in frail people" this result is interesting does not answer the objective of the study, review this information
In the course of the text, no hypothesis was presented, as if to affirm, as the researchers indicate, that the study hypothesis in which they refer that “the evident increase in the mortality rate among fragile patients supports the hypothesis of our study”. Likewise, the authors refer “Another possible cause could be related to the fact that people who are weaker and less likely to survive have a lower allocation of health resources than the subgroups with greater resistance to infection” but do not make the corresponding discussion, this could bring about a legal medical problem.
The authors do not present the limitations of the study

Conclusion
The conclusion does not answer the objective of the study. The following statement: "Higher frailty scores were associated with worse results and an increased mortality rate in ICU admissions" not identified among the results, it is suggested to review the information
What would be the recommendations? What would be the contribution of your work to the advancement of science and to the area of gerontology and geriatrics?

References
Suitable for the study

Annotated reviews are not available for download in order to protect the identity of reviewers who chose to remain anonymous.

·

Basic reporting

Minor comments:
1. Abstract:
-a sample of 254 patients”: specify in the abstract the main features of your patients (Age, if they are affected from COVID-19 or not)patients aged … with COVID-19.
-“after clinical”: add “features”.
-The word prognosis is non-specific and redundant to describe the outcome, as it already includes the outcomes previously reported(length of stay and ICU admission).
-The last part of the abstract can be implemented with data about: mean CFS score of frail and pre-frail patients, and HR of mortality among them, with p value to support your conclusive statements.
2. Main text:
-Line 54: change the acronym ICU with ICUs.
-Line 61: change systemic function with “systemic functional capacity”.
-Line 63: the message is unclear. I suggest you to write: “… and since it may affect both physical, physiological and cognitive functions, frailty itself demands implementation of a holistic approach to patient care”.
-Line 88: change “takes” with “take”.
-Line 110: specify how the diagnosis of SARS-CoV-2 infection was made.
- Line 133-137: enhance description of statistical methodology (see next section).
-Line 157: I suggest you to report once again CFS values used to identify robust, pre-frail and frail patients, in order to improve readability.
-Line 161-162: I suggest you to report summary statistics in support to your statements. You could say “The individuals classified as frail (CFS>=5) were mostly women (56.82%), and of older age (median[IQR]: 81.5 [72-87.75]), compared with the other groups”.
-Line 163: define the criterion used for diagnosing a nosocomial infection from SARS-CoV-2.
-Line 165: change “In” with “It”.
-Line 168: change chronic lung disease with “chronic bronchitis”, since also COPD would be included among chronic lung diseases, but it has not shown a similar linear increase in prevalence among groups.
-Line 169-170: the message of the statement “renal disease was remarkable..:” is not clearly expressed. You could say “. With the same pattern as heart disease, renal disease was less prevalent among robust patients (7.95%) compared to pre-frail and frail groups (20.95 % and 20.45%, respectively; p = 0.017).
-Line 174: I suggest you to improve the description of this part as “Differences in clinical presentation of the disease among the 3 groups were mostly not statistically significant; however, as shown in table 2, frail patients have shown the highest incidence of altered behaviour and level of consciousness (table 2).
-Line 182-184: report the incidence of major complications as percentage related to the overall population of patients included in the study.
-Minor punctuation mistakes: add periods at the end of the sentence at lines 59, 68, 78, 87.

Experimental design

Methodology of the study must be extensively revised, as it often not clearly reported.
1) Study variables: I suggest the authors to improve description of study variables included in the present study and to categorize them as follows:
-exposure variable: frailty as measured by CFS
-covariates: age, gender and other variables included in the study and known to potentially affect prognosis which are reported in table 2. Briefly describe which disease classification system have you referred to for diagnoses included in the study (e.g. ICD-9, 10)
-Outcome variables.

2) Statistical analysis: must be improved.
-Quantitative variables included in the study were mainly expressed in terms of median (and not mean) and IQR.
-Which statistical analysis have you performed to investigate the ability of frailty to predict length of stay and ICU admission, within the prospective design of your study? Have you used a logistic/linear regression model analysis?
-Which software for statistical analysis have you used for performing statistical analyses?
-Line 258: change hearth with heart.
-Line 279: briefly report data supporting the high mortality associated with delirium in frail people in this study (report HR or ‘data not shown’ within brackets).
-Line 287: add appropriate reference within brackets.

Validity of the findings

Study findings confirmed the role of frailty in guiding prognosis in hospitalized patients with COVID-19 and underlined the importance of using frailty assessment as a screening method for tailoring targeted interventions aimed at improving prognostic outcomes.
Discussion and conclusions are well stated and appropriately referenced; however, I suggest to describe the main limitations and points of strength of the present study in the final part of the discussion. Additionally, since multimorbidity and frailty may be bidirectionally related, you could discuss the role of a multidimensional approach integrating multimorbidity and frailty in the management of hospitalized patients.

Additional comments

This study investigated the interesting role of frailty in influencing outcomes of older patients hospitalized because of COVID-19. The descriptive part of the study is well written, with only minor revisions regarding English form, punctuation, and minor mistakes. However, the section about methods is too short and must be improved. Here above you find a point-by-point list of minor and major revisions that may be helpful to enhance the quality of the manuscript.

---

## Round 0.2 · Minor Revisions

Still pending some minor modifications suggested by one of the reviewers. Please, take them into account for the new version of the revised text.

·

Basic reporting

The authors have satisfactorily addressed all my concerns and made the necessary changes to the manuscript; all parts of the manuscript are now much more clearly presented and I think that results may help raise the importance of a multidimensional approach (including both frailty and multimorbidity) in the care of patients with COVID-19. I suggest some other minor revisions before final publication.

Experimental design

The methods were carefully described as well as presentation of study results.

Validity of the findings

In this retrospective cohort study, frailty was found to be an independent predictor of mortality among patients with COVID-19, as well as cardiovascular disease, hypertension, and chronic kidney disease. Using a multidimensional approach can improve care of complex elderly patients with COVID-19 and tailor targeted approach to counteract negative consequences according to disease severity, multimorbidity, and frailty.

Additional comments

The authors have satisfactorily addressed all my concerns and made the necessary changes to the manuscript; all parts of the manuscript are now much more clearly presented and I think that results may help raise the importance of a multidimensional approach (including both frailty and multimorbidity) in the care of patients with COVID-19. I suggest some other minor revisions before final publication.


Line 41: the hospital stay was similar between the 2 groups, thus I suggest removing the sentence “a shorter hospital stay (p=0.296) from the abstract.
Line 56: change “can be” with was.
Line 64: add 95 % CI after RR (1.24 to 2.09).
Line 90: add a comma after primary care setting.
Line 98: change period after “COVID-19 patients” with comma.
Line 118: change “does not take account” with “does not take into account”.
Line 124: change “worse prognosis; but pre-frail subjects, who have” with “worse prognosis, but pre-frail subjects as well, who have”.
Line 131: use periods instead of commas to separate the whole number from decimals.
Line 136: change “scores 1-3” with “Patients scoring 1-3”.
Discussion: introduce the important prognostic results in the first part of the discussion (e.g, “in this retrospective cohort study, we found that frailty was a major predictor of mortality among hospitalized elderly patients with COVID-19; cardiovascular disease, chronic kidney disease, and hypertension were also shown to predict decreased survival in the fully-adjusted model; frailty”.
Line 231: change “elderly, being a women, dependent” with “old age, female sex, dependency”.
After line 260: the terms “comorbidities” and “multimorbidity” should not be used interchangeably. I suggest you introduce the concept of “multimorbidity” which differs from simply comorbidity, since it does not give priority to any disease, but considers the whole number of multiple conditions and their interaction with functional status to finally determine patients’ outcomes”.

Table 4: I suggest reporting under the table which variables you included in the multivariable model.

Additionally: change all “pvalue” with “p value”.

---

## Round 0.3 · accepted · Accept

All the reviewers' concerns have been correctly addressed.

·

Basic reporting

The authors carefully addressed all my concerns. The paper is now suitable for publication in Peer J.

Experimental design

The methods were carefully described as well as the presentation of study results.

Validity of the findings

In this retrospective cohort study, frailty was found to be an independent predictor of mortality among patients with COVID-19, as well as cardiovascular disease, hypertension, and chronic kidney disease. Using a multidimensional approach can improve care of complex elderly patients with COVID-19 and tailor targeted approach to counteract negative consequences according to disease severity, multimorbidity, and frailty.

Additional comments

The authors have satisfactorily addressed all my concerns and made the necessary changes to the manuscript; all parts of the manuscript are now much more clearly presented and I think that results may help raise the importance of a multidimensional approach (including both frailty and multimorbidity) in the care of patients with COVID-19